# The Clinical and Genotypic Spectrum of Scoliosis in Multiple Pterygium Syndrome: A Case Series on 12 Children

**DOI:** 10.3390/genes12081220

**Published:** 2021-08-06

**Authors:** Noémi Dahan-Oliel, Klaus Dieterich, Frank Rauch, Ghalib Bardai, Taylor N. Blondell, Anxhela Gjyshi Gustafson, Reggie Hamdy, Xenia Latypova, Kamran Shazand, Philip F. Giampietro, Harold van Bosse

**Affiliations:** 1Shriners Hospitals for Children, Montreal, QC H4A 0A9, Canada; frauch@shriners.mcgill.ca (F.R.); GBardai@shriners.mcgill.ca (G.B.); rhamdy@shriners.mcgill.ca (R.H.); 2Faculty of Medicine and Health Sciences, McGill University, Montreal, QC H3G 2M1, Canada; 3Inserm, U1216, Grenoble Institut Neurosciences, Génétique médicale, Université Grenoble Alpes, CHU Grenoble Alpes, 38000 Grenoble, France; KDieterich@chu-grenoble.fr (K.D.); XMartin@chu-grenoble.fr (X.L.); 4Shriners Hospitals for Children, Philadelphia, PA 19140, USA; tblondell@shrinenet.org; 5Shriners Hospitals for Children Headquarters, Tampa, FL 33607, USA; agustafson@shrinenet.org (A.G.G.); kshazand@shrinenet.org (K.S.); 6Pediatric Genetics, University of Illinois, Chicago, IL 60612, USA; philipg@uic.edu

**Keywords:** *CHRNG*, distal arthrogryposis type 8, Escobar, multiple pterygium syndrome, *MYH3*, scoliosis

## Abstract

Background: Multiple pterygium syndrome (MPS) is a genetically heterogeneous rare form of arthrogryposis multiplex congenita characterized by joint contractures and webbing or pterygia, as well as distinctive facial features related to diminished fetal movement. It is divided into prenatally lethal (LMPS, MIM253290) and nonlethal (Escobar variant MPS, MIM 265000) types. Developmental spine deformities are common, may present early and progress rapidly, requiring regular fo llow-up and orthopedic management. Methods: Retrospective chart review and prospective data collection were conducted at three hospital centers. Molecular diagnosis was confirmed with whole exome or whole genome sequencing. Results: This case series describes the clinical features and scoliosis treatment on 12 patients from 11 unrelated families. A molecular diagnosis was confirmed in seven; two with *MYH3* variants and five with *CHRNG*. Scoliosis was present in all but our youngest patient. The remaining 11 patients spanned the spectrum between mild (curve ≤ 25°) and malignant scoliosis (≥50° curve before 4 years of age); the two patients with *MYH3* mutations presented with malignant scoliosis. Bracing and serial spine casting appear to be beneficial for a few years; non-fusion spinal instrumentation may be needed to modulate more severe curves during growth and spontaneous spine fusions may occur in those cases. Conclusions: Molecular diagnosis and careful monitoring of the spine is needed in children with MPS.

## 1. Introduction

The term pterygium is used most commonly to describe an acquired ophthalmologic condition, with a conjunctival “wing” or flap that can cross the cornea. The term also describes joints with congenital webbing or winging of soft tissue which limits the joint motion. Usually, more than one joint and body region is involved, meeting the criteria for arthrogryposis multiplex congenita [1]. While pterygia may be an incidental finding in persons with arthrogryposis, such as the occasional knee pterygium found associated with severe contractures in Amyoplasia, they can also manifest as a more generalized syndrome, such as popliteal pterygium syndrome or multiple pterygium syndrome (MPS).

MPS is a rare form of arthrogryposis multiplex characterized by a constellation of congenital anomalies [2]. The webbing of skin and contractures of the joints that are found in this disorder may restrict movement. Examples of joint involvement in the upper extremities include axillary pterygia (both the anterior and posterior folds), elbow flexion contractures with antecubital webbing, mildly dorsiflexed wrists, and fingers with camptodactyly, interdigital pterygia and thumb-in-palm deformities. In the lower extremities, perineal pterygia can span medially from one thigh to the other, knee flexion contractures with pterygia can be severe, and foot deformities include both clubfoot and congenital vertical talus (rocker bottom foot).

Other characteristic findings of MPS are short stature, webbing of the neck, and distinctive facial features including micrognathia, cleft palate, down-turned corners of the mouth, an elongated philtrum, down-slanting palpebral fissures, epicanthal folds, ptosis, and low-set ears, all of which are related to diminished fetal movement. Developmental spine deformities are common, not only as a coronal plane deformity (scoliosis), but frequently with a substantial associated sagittal plane deformity, making it a kyphoscoliosis. Spontaneous spinal fusion abnormalities occur often in MPS and are congenital.

MPS can be separated into the lethal pterygium syndromes (LMPS, MIM253290) and the non-lethal syndromes; the latter conditions are categorically referred to as Escobar syndrome (or Escobar variant MPS, MIM265000) and most are autosomal recessive. Here we will only be discussing the non-lethal forms of MPS.

Several genes associated with MPS give rise to what has recently been described as a prenatal form of myasthenia, first associated with variants in *CHRNG*. *CHRNG* codes for the γ subunit of the acetylcholine receptor (AChR) in the developing fetus. Mutations that impact the expression of this subunit, its integration in the AChR, or the transport of the AChR to the sarcolemma will have major consequences to the neuromuscular junction. At 33 weeks gestation though, the γ subunit starts to be replaced by the ɛ subunit of AChR, finally leading to a functional adult neuromuscular junction. However, effects of the fetal akinesia have become manifest before that stage of development and the damage is irreversible. Interestingly, individuals with Escobar syndrome do not have muscle weakness or electrophysiological symptoms associated with myasthenia gravis postnatally, as the AChR essentially functions normally after birth. Other causes of Escobar syndrome recessive gene mutations of the AChR include *CHRNA1* (α1-AChR subunit), *CHRNB1* (β1-subunit), *CHRND* (δ-subunit), and *RAPSN* (AChR binding protein). Escobar syndrome can also be caused by pathogenic variants in *CNTN1* (contactin 1) and *DOCK7* (dedicator of cytokinesis 7) [3,4,5]. See Figure 1.

More recently, cases of MPS with an autosomal dominant form, also classified as distal arthrogryposis type 8 (DA8) [6,7], have been associated with mutations in the embryonic myosin heavy chain gene, *MYH3*. This is the same gene that is found underlying other forms of distal arthrogryposis (DA1, DA2A or Freeman Sheldon syndrome, and DA2B or Sheldon Hall syndrome).

Spinal curvatures in Escobar syndrome appear early, often present at birth, and can progress quickly. Treatment options include spine casting, bracing, and expandable implant surgery, to allow as much chest growth and development as possible, and ultimately spinal fusion, with a goal of well-balanced spine [8]. Although advances in the phenotypic spectrum, disease progression and genetic etiology of MPS have been made [9,10], definite phenotype–genotype correlations need yet to be discovered. The objective of this case series was to describe the phenotypic presentations of a small multisite collection of patients with Escobar syndrome, in particular as to how it relates to their spinal deformity. By determining the genetic basis of their MPS, we hope to associate the genotype with their phenotype and natural history of their scoliosis, and describe the clinical interventions aimed at reducing the spinal curvatures associated with these cases.

## 2. Materials and Methods

### 2.1. Design

Case series using a retrospective chart review and prospective data collection at three hospital centers; Shriners Hospitals for Children in Montreal, Canada, Shriners Hospitals for Children in Philadelphia, United States, and Grenoble Alpes University Hospital Center in France. The patients at the two Shriners hospitals were enrolled in a multisite pediatric arthrogryposis registry that collects clinical, patient-reported outcomes, and provides whole genome sequencing to map the phenotype to genotype in this population.

### 2.2. Ethical Approval and Consent

Written informed consent according to local ethical committees in all participating centers was obtained for all patients. Clinical and genetic data were anonymized and entered into a secured database accessible only to the research team.

### 2.3. Patients

We describe the phenotypic presentation of 12 patients presenting with MPS and scoliosis. Clinical data were analyzed retrospectively. Clinical data (sex, age), scoliosis (i.e., age at onset, congenital/early onset/late onset), and a description of the phenotype was extracted from the electronic medical record and evaluated in clinic (contractures, limb anomalies, webbing/pterygia, other system involvement). A description of interventions, including type (observation, bracing, and surgery) and age at treatment was collected. Spine X-rays were reviewed and classified in order to better understand the curves in children with Escobar syndrome using the following classification system of curve behavior. A mild curve was defined as a curve ≤ 25° at any age. A moderate curve was a curve > 25° but ≤50° at any age. Severe curves exceeded 50°, but not until after 4 years of age, whereas malignant curves were ≥50° before 4 years of age.

### 2.4. Genotyping and Data Analysis

Genomic DNA was isolated from saliva collected in Oragene OGR-600 tubes (DNA Genotek, Kanata, ON, Canada) according to manufacturer’s instructions and subsequently extracted using the Chemagic360 platform (PerkinElmer, Waltham, MA, USA). DNA was quantified using Qubit. Average gDNA obtained from extractions was 40 ng/uL. Libraries were prepared in the Zephyr G3 NGS Automated Workstation (PerkinElmer) using the Illumina DNA PCR-Free Prep, Tagmentation (Illumina, San Diego, CA, USA) with 1000 ng of input gDNA. Resulting libraries were quantified prior to pooling using the Kapa Library Quantification Kit (Roche, Basel, Switzerland). Libraries were pooled at a concentration of 1.4 nM and were sequenced on the NovaSeq 6000 sequencing platform (Illumina). Paired-end sequencing (2 × 150 bp) was performed at >30× coverage per sample. GRCh38 reference genome was used. Genotyping at the Grenoble arthrogryposis reference center was conducted using DNA extracted from whole blood samples collected on EDTA tubes. Libraries were prepared with the Nextera DNA Flex Library Prep Kit (Illumina). Exome sequencing was performed on a NextSeq 500 (Illumina).

Data analysis, including variant calling, was performed using the Illumina TruSight^™^ Software Suite platform. This platform performs alignment, variant calling, variant annotation and filtering. For the Grenoble hospital center, bioinformatic analysis was performed with an in-house pipeline using an NGS platform, using the Burrows-Wheeler Aligner, Picard Tools 2.18.23-0 and GATK v4.0.12. Variants were interpreted with ANNOVAR and prioritized through python annotation scripts. The strategy for genome data interpretation was primarily based on disease and phenotype gene target definition. Data were descriptively analyzed.

## 3. Results

### 3.1. Clinical Features

The 12 patients (7 females) with MPS and scoliosis were from 11 unrelated non-consanguineous families. They were born in Algeria (*n* = 1), Canada (*n* = 1), France (*n* = 4) and the United States (*n* = 6) and were between 1 and 20 years old at the time of last follow up (Table 1 and Table 2). Family history was positive in one family (F6), with presence of clinical features of Escobar and scoliosis in the mother and maternal aunt of the index patient (P7). In the family of two affected siblings (F2), one pregnancy had been terminated due to severe fetal deformities; muscle histology in that fetus was in accordance with a diagnosis of Escobar syndrome.

The children were born after 34–41 weeks of pregnancy (Table 3). Six of the children were born vaginally, while the other six were born via c-section due to breech position and complicated pregnancy with oligohydramnios or polyhydramnios. Eight children were admitted to the neonatal intensive care unit after birth due to feeding difficulties and/or respiratory issues.

Characteristic phenotypic features of Escobar were observed in all 12 patients, including webbing over the neck, axillary, elbow, knee and/or fingers (Table 1). Downslanting palpebral fissures were observed in 10 patients (Table 2). Three patients had cleft palate and another five had high palate. Four patients had both posterior rotated ears and low-set ears, whereas another three patients either had posterior rotated ears or low-set ears. Figure 2 and Figure 3 showcase pterygia and characteristic features in MPS among two patients.

Seven of the 12 patients had foot deformities, six with congenital vertical tali and one with clubfeet (Table 1). Contractures were widespread across several joints (e.g., shoulders, elbows, wrists, fingers, hips, knees, ankles) and varied in severity among patients. Regarding functional mobility (data available in 11 patients), five patients were independent walkers, one used a stroller as she did not crawl or walk yet at age 22 months, two used a manual wheelchair for outdoor mobility, and three used a motorized chair for outdoor mobility. Of the five children who ambulated without a mobility aid, four wore knee ankle foot orthoses and two walked with a crouched gait.

Scoliosis was present in all but one patient who was only nine months of age at the last assessment. Four patients had malignant curves that had developed either during infancy or possibly prenatally (prenatal scoliosis). The level of the major curve was at the thoracolumbar spine in two of these patients, the other two patients had left low thoracic curves. These curves all included the pelvis, causing moderate to severe pelvic obliquity (18–50°). One patient had a mild but sweeping kyphosis including the pelvis, two had apical mid thoracic hyperkyphosis, whereas the last had thoracic hypokyphosis but with a thoracolumbar kyphosis. All four patients with malignant curves between 70° and 120° underwent non-fusion spinal instrumentation at 2, 4, 7 and 10 years of age in which expandable implants were placed to help control the curve during growth. Spine X-rays show a malignant curve in Figure 4.

Severe curves were noted in three patients, with curves progressing more slowly than the malignant curves, so that surgery could be delayed. Two patients had radiographic findings of curves at 3 and 11 months of age, whereas the third had periodic radiographs which did not demonstrate a curve until 6 years of age. All three patients had a right thoracic curve pattern, with two having thoracic hypokyphosis, which was confluent with the lumbar lordosis, the third had a thoracolumbar kyphosis. One patient was undergoing brace treatment, another had a non-fusion spinal instrumentation procedure at 11 years of age, and the last had a formal spinal fusion at 12 years of age (Figure 5).

Moderate curves were found in two patients, both detected during the first year of life. Both were left sided curves, one a thoracolumbar curve with hyperkyphosis and hypolordosis, the other a lumbar curve with thoracic hypokyphosis. One patient underwent serial casting at 2 years of age, while the curve of the other did not progress past 25° until 12 years of age, and therefore was not treated. Two patients, 5 and 8 years of age at last follow up, had mild curves. They both had a forward lean on the standing sagittal radiographs with flexed hips and mild lumbar hyperlordosis. Figure 6 shows spine X-rays of a child with a moderate curve.

Thus, a total of five patients had undergone a non-fusion spinal instrumentation procedure. Interestingly, post-operative follow-up demonstrated that each of these patients had spontaneous fusions of vertebral levels, also called autofusions, in at least one uninstrumented vertebral interspace, depicted in Figure 7.

### 3.2. Molecular Analysis

Whole genome or whole exome sequencing was performed in 8 of the 12 patients and their parents, when available. Parents were clinically unaffected, except in family F6, as described earlier. Pathogenic recessive variants in *CHRNG* were found in five patients, all of whom had compound heterozygous variants (Table 4). Among the five different *CHRNG* variants found in these individuals, four led to premature termination codons and one represented an in-frame duplication (p.Trp98_Leu100dup) that had been described before [11]. According to family history, one other patient (P4) had compound heterozygous *CHRNG* mutations, but no detailed information was available and no DNA sample could be obtained.

Pathogenic variants in *MYH3* were found in two patients. One patient had a dominant de novo missense variant (p.Leu1204Pro), which affects the tail region of *MYH3*. This variant is not present in gnomAD and is predicted to be pathogenic by nine different prediction algorithms. Two other missense variants leading to substitutions of amino acids in the *MYH3* tail domain by proline residues have been described in the literature [12]. The other patient with pathogenic *MYH3* variants was compound heterozygous for a known recurrent splice variant (c.-9+1G > A) [12] and a novel missense variant (p.Ala183Pro). This missense is not present in gnomAD and affects the head region of MYH3, a domain where pathogenic missense variants are frequently observed [12].

A number of novel variants in genes reported to be associated with AMC were identified in patient P7, whose mother and maternal aunt have features of Escobar and scoliosis. Further validation of these variants is required given the novelty, and thus were not reported in this paper.

## 4. Discussion

Here we describe 12 individuals with variable pterygia, mild to severe flexion contractures of several joints and spine anomalies. In six of the patients, the disorder was caused by biallelic *CHRNG* mutations, one patient had an apparently dominant de novo mutation in *MYH3* and one patient was compound heterozygous for *MYH3* variants. Scoliosis was highly prevalent in our patient cohort and extremely severe in some. Intriguingly, all patients undergoing non-fusion spinal instrumentation subsequently developed fusion in at least one vertebral segment that had not been directly touched by the surgical intervention.

Biallelic loss of function variants in *CHRNG* are a well-established cause of Escobar syndrome [3,4]. Similar to previous studies, we observed that the recurrent frameshift c.459dup mutations in several unrelated patients [9]. In addition, we found that one of our patients had compound heterozygous *MYH3* variants, including the previously characterized splice variant c.-9+1G > A in the 5′ untranslated region of the gene [12,13]. Interestingly the MYH3 c.-9+1G > A variant was initially described in individuals with spondylocarpotarsal synostosis syndrome, but the reported phenotype also included webbing, contractures and scoliosis [12] and thus had considerable overlap with the clinical characteristics of our patients. Moreover, *MYH3* mutations are associated with distal arthrogryposis type 8. Based on recommendations by Biesecker et al. [14] it might be useful to use the broad term *MYH3*-related disorder to encompass these different conditions. 

Multiple pterygia are seen in many types of lethal forms of AMC [15,16,17,18], but only three genes have been associated with non-lethal MPS or Escobar syndrome, namely *CHRNG*, *MYH3*, and *TPM2*. One explanation might be the exclusive or predominant expression of these genes during development and their lack or diminished expression postnatally, whereas other genes seem to be pre- and postnatally expressed. Indeed, the *CHRNG* encoded gamma-subunit of AchR, as a developmental subunit, stops being expressed at the end of the second and start of the third trimester [19], a timepoint from which the postnatal/adult CHRNE-encoded epsilon subunit is incorporated into the AchR. That most certainly also explains why individuals with *CHRNG*-related MPS do not present with clinical or electromyographic myasthenic symptoms after birth. In the same line, the *MYH3* encoded embryonic myosin heavy chain, despite its role on fiber type, fiber number, and muscle fiber differentiation [20], is much less expressed in human postnatal muscle [21]. This holds also true for *TPM2* [22]. 

The development of multiple pterygia probably reflects the spectrum of the most severe consequence and earliest onset of fetal akinesia. In this regard, pathogenic variants in *CHRNG*, *MYH3* and *TPM2* described to date have been associated with loss of function mutations. In *CHRNG*, nonsense, splice site and missense mutations alike have been shown to abolish AchR expression at the sarcolemma [3,4], thus probably completely abolishing neuromuscular transmission during embryonic and fetal development. Of note, the AchR is still expressed even in case of loss of function mutations in other genes involved in the maintenance and function of the neuromuscular junction, such as *RAPSN* or *MUSK* [9,23,24], leading to a severe fetal akinesia deformation sequence phenotype, but without pterygia.

Scoliosis in children with Escobar syndrome is extremely prevalent, with most curves eventually progressing to treatment. Other authors have reported prevalence rates of scoliosis in patients with multiple pterygium syndrome between 32% and 93% [25,26,27,28]. Of the 12 patients covered in this report, only three did not yet require any scoliosis treatment, one of whom is only 9 months old, the other two are under 10 years. Four of the patients had a spine deformity that behaved beyond what is typically considered severe, therefore we named curves that were greater than 50° before 4 years of age “malignant”. The patients with malignant curves also had more severe limb involvement, most commonly including severe hip and knee flexion contractures with pterygium. Most malignant curves were first detected in infancy, some even in the neonatal time period, suggesting that some were actually prenatal curves, curves that reflected the patients’ unchanging intrauterine position due to severe fetal akinesia. Importantly, other patients also had curves detected during their first year of life, but did not develop malignant curves, varying between mild and severe. The malignant curves all had pelvic obliquity near or greater than 20°, an indication of the uncompensated nature of these curves. Two of the patients in the severe category initially had their curves detected before their first birthday, but the third patient (Patient 3) was routinely monitored with serial radiographs, and no curve was detected prior to 6 years of age. All three of patients with severe curves underwent bracing of their spines, which allowed further growth prior to needing surgical stabilization of their curves. One of the shortcomings of the classification system we used is that a patient may progress from one class to another as they grow and their curves progress, and that the classification was based on a very small group of subjects. It is likely that some of the mild and moderate curves will progress to severe prior to the patient reaching skeletal maturity. We felt, though, that it provided some structure with which to analyze and group the subjects.

We expect that larger studies will provide insights, allowing for improvements of the scoliosis classification system used in this manuscript. A molecular diagnosis was confirmed in seven of our patients, two of which were found to have a *MYH3* gene mutation; the remaining five had a *CHRNG* mutation. The two patients with *MYH3* mutations both had malignant curves. The two other patients with malignant curves both had *CHRNG* gene mutations, of which one, Patient 2, was the older sibling of Patient 3, with the exact same mutation, but Patient 3 had a severe scoliosis. We have two other sibling pairs that we could not include in this study, since they did not have complete data available. One brother–sister pair both had malignant curves that started in infancy, whereas the brother of the other brother–sister pair had a malignant curve that started in infancy yet his sister’s curve did not become severe (crossing 50°) until after 10 years of age. Curve patterns varied modestly in our cohort. Two of the malignant curves were thoracolumbar extending to the pelvis, one left-sided the other right, whereas the other two were low thoracic apex curves, also extending nearly to the pelvis, both left-sided. Three of the spines had thoracic hyperkyphosis, but the fourth had a lordotic thoracic spine with a mild thoracolumbar kyphosis. The severe curves were right sided mid- or lower thoracic-apex curves, with relatively lordotic thoracic spines, two patients had a mild thoracolumbar kyphosis. The mild and moderate curves in general were a combination of right thoracic and/or left lumbar curves, and usually a lumbar hyperlordosis. This variability complements the findings of Margalit and colleagues [29], who found three of their nine patients had right-sided thoracolumbar curves, and the rest had left-sided thoracolumbar curves. They did not describe the patterns of sagittal appearance of their patients. Coalescence of vertebral levels the spine appears to be common in patients with Escobar syndrome, particularly in the severe and malignant curves. Margalit et al. [29] noted the same on pre-operative computer tomography (CT) of their patients.

In our cohort, it is clear that spontaneous vertebral fusions occur, particularly in the severe and malignant curves, best seen in the children undergoing non-fusion spinal instrumentations. In these spines, progressive intervertebral fusions are seen both anteriorly and posteriorly in the uninstrumented section of the spine, suggesting either that lack of intervertebral motion, or the distraction of the space, leads to the fusion. We did not identify vertebral abnormalities or lack of vertebral segmentation in the films of our patients under 2 years of age, although we did not have CT scans and details could be difficult to visualize on the films. Therefore, we were unable to resolve if any of the intervertebral fusions were congenital failure of segmentation, but we suspect that most, if not all, were due to postnatal spontaneous fusions. Clearly, patients with Escobar syndrome need to be carefully monitored for the development of scoliosis, and aggressively treated to postpone the need for surgical intervention. Joo et al. [28] noted a tethered cord or a syrinx in 4 of their 16 patients. Although only one of our patients needed detethering of their spinal cord (Patient 5), treating physicians need to be vigilant for such possibilities. Both bracing and serial spine casting appear to be beneficial to some extent in controlling the curve and allowing further growth for at least a few years. Patients with spine-induced pelvic obliquity, particularly those apparent in infancy, likely have malignant curves. This seems to be particularly true for patients with *MYH3* mutations underlying their Escobar syndrome. Parents need to be informed about the challenging nature of the curve, and that non-fusion spinal instrumentation will be needed to try to modulate the curve during growth. Patients undergoing non-fusion spinal instrumentation are likely to experience spontaneous fusions of their spine, which may limit the amount of expansion possible during the child’s growing years. Spine balance must be a priority at the initial implantation of the expandable device, as a formal fusion may not be necessary due to the spontaneous fusions, so long as spine balance is satisfactory. Conversely, if the spine is not well balanced, a formal fusion after a non-fusion spinal instrumentation will be very challenging due to the fusions.

## 5. Conclusions

In conclusion, we found a unique spine phenotype in these patients with MPS caused by *CHRNG* and *MYH3* mutations. More detailed characterization using 3D imaging may help further refine this spine phenotype in patients with MPS.

## Figures and Tables

**Figure 1 genes-12-01220-f001:**
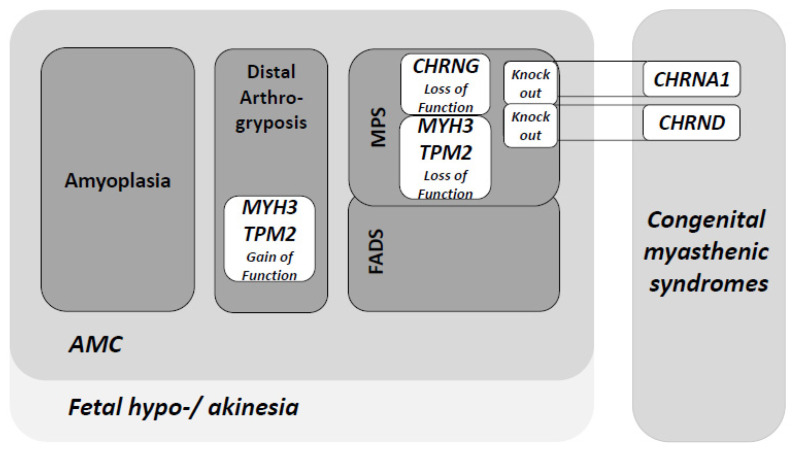
Clinical spectrum and overlap of molecular causes of arthrogryposis multiplex congenita (AMC). The main categories of AMC phenotypes are represented as dark grey boxes. Some of the genetic determinants that are responsible for multiple phenotypes among AMC subtypes are represented in white boxes. Both *MYH3* and *TPM2* biallelic loss of function have been associated to MPS, although dominant heterozygous variants of these genes are linked to several forms of distal arthrogryposis. FADS: fetal akinesia deformation sequence; MPS: multiple pterygium syndrome.

**Figure 2 genes-12-01220-f002:**
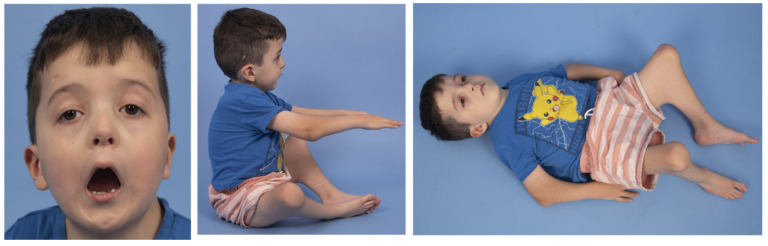
Patient 1 with micrognathia, downslanting palpebral fissures, lowset and posteriorly rotated ears, and pterygia apparent in the antecubital and knee popliteal regions.

**Figure 3 genes-12-01220-f003:**
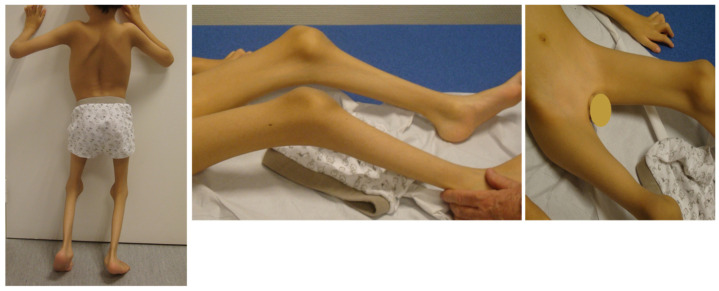
Patient 10 with mild to moderate axillary, popliteal, and inguinal pterygia.

**Figure 4 genes-12-01220-f004:**
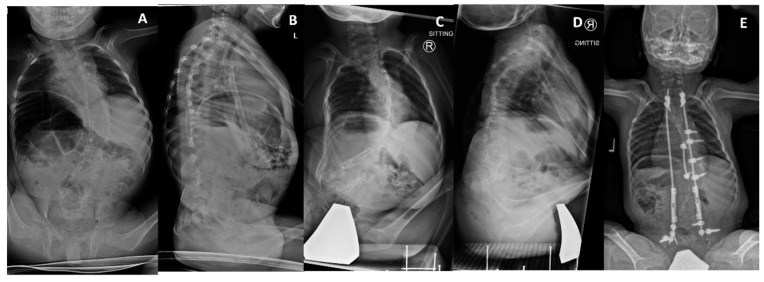
Patient 1 has compound heterozygous pathogenic *MYH3* variants. Curve was first noted at 13 months of age. (**A**,**B**) Posterior-anterior and lateral spine radiographs at 34 months of age showing a 68° curve with mild pelvic obliquity, and mild thoracolumbar kyphosis. (**C**) At 5 years old, the curve is 94°, with 33° pelvic obliquity and (**D**) more pronounced low thoracic kyphosis extending into the lumbar spine. (**E**) Latest follow-up at 7 years old, following halo gravity traction and a non-fusion spinal instrumentation at age 5.

**Figure 5 genes-12-01220-f005:**
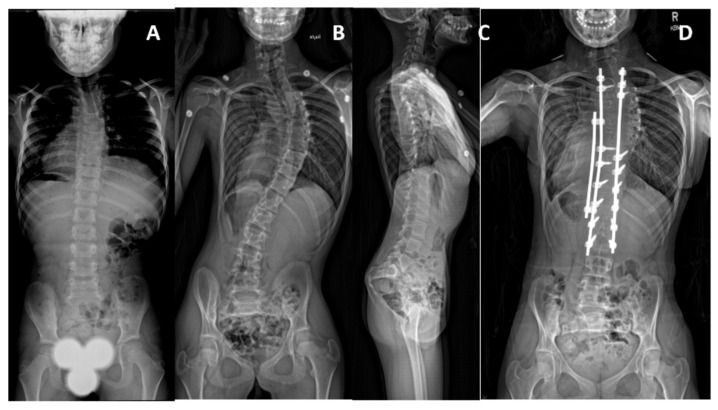
Patient 3 has compound heterozygous *CHRNG* variants and severe curve behavior. Curve first noted at 6 years of age. (**A**) Standing posterior-anterior spine films at 6 years old demonstrate a 19° right-sided thoracic curve. (**B**,**C**) A 71° curve with a lordosis spanning the thoracic and lumbar spine. (**D**) Spinal fusion at 13 years of age.

**Figure 6 genes-12-01220-f006:**
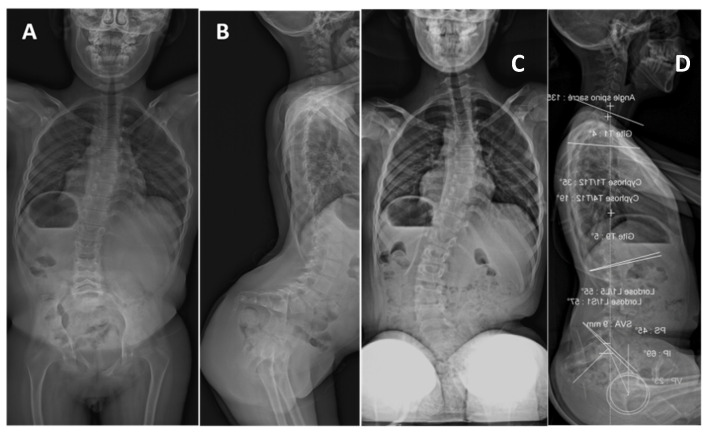
Patient 10 has compound heterozygous *CHRNG* variants, and a moderate scoliosis, first identified during infancy. (**A**) Posterior-anterior supine film at 10 years old, demonstrating 15° right thoracic and 18° left lumbar curves, and lateral view (**B**) Showing mild thoracic hypokyphosis and hip flexion contractures driving lumbar hyperlordosis. At this size, the curves would be considered mild. (**C**,**D**) Posterior-anterior and lateral sitting spine films at 12 years of age, showing that the curve had mildly progressed to 28°, reaching criteria for a moderate curve. The sagittal profile continues to show mild thoracic hypokyphosis.

**Figure 7 genes-12-01220-f007:**
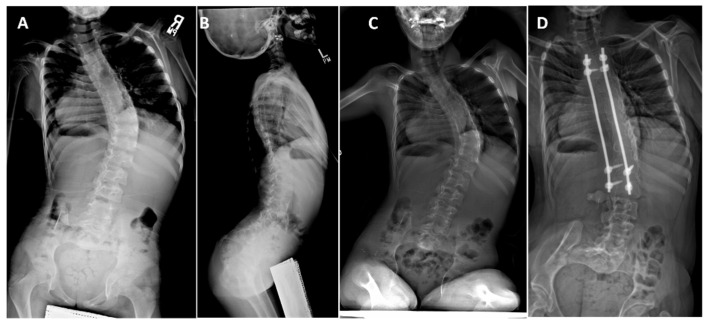
Patient 5 with compound heterozygous *CHRNG* variants, and severe scoliosis, first identified at 3 months of age. Spine bracing started at 24 months old. (**A**) Posterior-anterior and lateral standing films at 7 years old demonstrate a 53° right low thoracic curve, with (**B**) thoracic and lumbar lordosis on the lateral film, and flexed hips indicating hip flexion contractures. (**C**) Sitting spine films at 11 years old with curve at 66°, before non-fusion spinal instrumentation. (**D**) Autofusion along the length of the instrumented spine is seen 3 years after rod replacement at 17 years of age.

**Table 1 genes-12-01220-t001:** Phenotype: pterygia and joints affected.

Patient	Family	Sex	Age(Years)	Multiple Pterygia (HP: 0001059)	Neck Pterygia (HP: 0009759)	Axillary Pterygia (HP: 0001040)	Pterygia—Other Locations	Scoliosis (HP: 0002650)	Micrognathia (HP: 0000347) and/or Retrognathia (HP: 0000278)	Talipes Equinovarus (HP: 0001762)	Congenital Vertical Talus (HP: 0001838)
P1	F1	M	7	+	+	+	Elbows, knees, digits	+	+	-	-
P2	F2	M	20	+	+	-	Knees, elbows, digits	+	-	-	+
P3	F2	F	13	+	-	-	Knees	+	-	-	-
P4	F3	F	10	+	+	+	-	+	-	-	+
P5	F4	F	18	+	-	-	Elbows, knees	+	+	-	+
P6	F5	F	2	+	+	-	Elbows, knees	+	+	-	+
P7	F6	F	8	+	+	+	-	+	+	+	-
P8	F7	M	11	+	+	+	Hips, knees	+	+	-	-
P9	F8	F	1	+	+	+	Elbows	-	+	-	+
P10	F9	M	13	+	-	+	Hips, knees	+	+	-	-
P11	F10	F	8	+	+	+	Digits	+	+	-	+
P12	F11	M	6	+	+	+	Digits	-	+	-	-

**Table 2 genes-12-01220-t002:** Phenotype: facial features and other characteristics.

Patient	Downslanting Palpebral Fissures (HP: 0000494)	Posteriorly Rotated Ears (HP: 0000358)	Low Set Ears (HP: 0000369)	Cleft Palate (HP: 0000175)	High Palate (HP: 0000218)	Other
P1	+	+	+	-	-	High nasal bridge (HP:0000426); hydrocele; inguinal hernia; tongue tie which was retracted and cauterized to divide frenulum
P2	+	-	-	-	+	Trismus, clenched hands (HP:0001188); small mouth; fusion of posterior elements of C2 and C3
P3	-	-	-	-	-	Extra cervical rib (HP:000089)
P4	+	+	+	-	+	Laryngomalacia (HP:00000060); Sprengel deformity (HP:0003745)
P5	+	-	+	-	+	Laryngomalacia (HP:00000060); corkscrew esophagus; tethering of spinal cord at age 1
P6	+	+	-	-	-	Band around right distal thigh; hypoglycemia at birth; hypoplasia of occipital lobes with prominent subarachnoid spaces
P7	-	-	-	+	-	-
P8	+	-	-	-	-	Ptosis (HP:0000508); convergent strabismus
P9	+	+	+	-	+	Facial hemangioma (HP:0000329); postnatal growth restriction
P10	+	-	+	-	+	Tongue atrophy (HP:0012473); ptosis (HP:0000508); postnatal growth restriction
P11	+	+	+	+	-	Tongue atrophy (HP:0012473)
P12	+	-	-	+	-	Right pneumothorax at birth

**Table 3 genes-12-01220-t003:** Pre- and postnatal information.

Patient	Prenatal Detection				Delivery		Postnatal		
	Oligohydramnios	Polyhydramnios	Lack of Fetal Movement/Contractures	Other Findings	GA (Weeks)	Type	Feeding Difficulty	Intubation	NICU
P1	-	-	+	Initial hydrops fetalis which resolved	38	c-section	+	-	+
P2	-	+	+	Fluid in right lung which resolved, right diaphragmatic plication at 2 weeks of age	41.5	Vaginal	+	-	+
P3	-	-	-		40	Vaginal	-	-	+
P4	+	-	+	Initial clinical impression of trisomy 18	34.5	c-section	+	+	+
P5	-	+	+		34	c-section	+	+	+
P6	-	+	+		39	c-section	+	-	+
P7	-	-	-		34	c-section	+	-	+
P8	-	-	+	Cystic hygroma at 12 weeks of gestation; vertebral block at 22 weeks of gestation; IUGR	38	Vaginal	+	-	-
P9	-	-	-	IUGR	41	Vaginal	-	-	-
P10	-	-	-	IUGR	37	Vaginal	-	-	-
P11	-	-	-		38	Vaginal	+	-	-
P12	+	-	-		37	c-section	+	-	+

IUGR: intra uterine growth restriction; NICU: neonatal intensive care unit.

**Table 4 genes-12-01220-t004:** Genotype information.

Patient	Family	Test	Gene	Variant 1	Variant 2	Inheritance
P1	F1	WES/ WGS	*MYH3*	c.-9+1G > A	c.547G > C p.Ala183Pro	AR
P2	F2	WGS	*CHRNG*	c.459dupA p.Val154SerfsTer24	c.753_754delCT p.Val253AlafsTer44	AR
P3	F2	WGS	*CHRNG*	c.459dupA p.Val154SerfsTer24	c.753_754delCT p.Val253AlafsTer44	AR
P4	F3	-	*CHRNG* *	Unavailable	Unavailable	AR
P5	F4	WGS	*CHRNG*	c.459dupA p.Val154SerfsTer24	c.401_402delCT p.Pro134ArgfsTer43	AR
P6	F5	WGS	*CHRNG*	c.459dupA p.Val154SerfsTer24	c.639_643delCAAGA p.Lys214AlafsTer82	AR
P7	F6	WGS		Negative		Dominant
P8	F7	AMC gene panel	*MYH3*	c.3611T > C p.Leu1204Pro	-	De novo
P9	F8	Single gene sequencing	*CHRNG*	c.202C > T p.Arg68Ter	c.292_300dup p.Trp98_Leu100dup	AR
P10	F9	WES		Negative		Unknown
P11	F10	WES		Pending		Unknown
P12	F11	WES		Pending		Unknown

WES: whole exome sequencing; WGS: whole genome sequencing. * According to family history, detailed information not available.

## Data Availability

The data presented in this study are available on request from the corresponding authors. The data are not publicly available due to the ethical and privacy nature of the data.

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
