# Peer review of "The Clinical and Genotypic Spectrum of Scoliosis in Multiple Pterygium Syndrome: A Case Series on 12 Children"

_genes, 2021, doi:10.3390/genes12081220_

Round 1

Reviewer 1 Report

The authors conducted a retrospective chart review and prospective data collection of multiple pterygium syndrome patients in three hospital centers. They described the clinical features and scoliosis treatment on 12 patients from 11 unrelated families.  The whole exome or whole genome sequencing was conducted and 2 patients with MYH3 variants and 5 patients with CHRNG were found. These data strongly support that MYH3 and CHRNG mutation causes multiple pterygium syndrome in humans. The work is compelling and interesting, the data is clear, and the discussions are adequate.

I would like to give one suggestion:

  • For patient 7 in Table 1, What is the mean of “?” ? If the data of patient is not available, please replace "?" with NA (not available). Patient 6 in Table 2 has the same problem.

Author Response

Thank you for your comments. We have received the missing data for patient 7 and have added the information to tables 1 and 2.

Reviewer 2 Report

This case series describes the clinical features and scoliosis treatment on 12 patients from 11 unrelated multiple pterygium syndrome (MPS) families. Retrospective chart review and prospective data collection were conducted at three hospital centers. A molecular diagnosis was confirmed in seven; two with MYH3 variants and five with CHRNG variants. Scoliosis was present in all but one patient.

The data presented here are descriptive, well-presented, and well-written and provide additional insight into the association of scoliosis with MPS. 

I have only minor comments.

Lines 74-75, the subunits of the acetylcholine receptor in parentheses are not readable.

Lines 157-9, this information should be removed: “This section may be divided by subheadings. It should provide a concise and precise description of the experimental results, their interpretation, as well as the experimental conclusions that can be drawn.”

Table 1. HP numbers do not seem necessary. Indicate in the legend for example: “HPO terms were used when possible”. Also, in the column "age" indicate "(years)" in brackets. For the youngest patients indicate the age in months.

A table including clinical and molecular information would be useful, along the lines of Table 1 in this article. https://www.sciencedirect.com/science/article/pii/S000292971830140X?via%3Dihub

Line 336: The acronym “FADS” should be explained.  

Author Response

Thank you for your revision and suggestions! we have addressed your revisions as follows:

Lines 74-75, the subunits of the acetylcholine receptor in parentheses are not readable.

response: this has been rectified.

Lines 157-9, this information should be removed: “This section may be divided by subheadings. It should provide a concise and precise description of the experimental results, their interpretation, as well as the experimental conclusions that can be drawn.”

response: this has been rectified.

Table 1. HP numbers do not seem necessary. Indicate in the legend for example: “HPO terms were used when possible”. Also, in the column "age" indicate "(years)" in brackets. For the youngest patients indicate the age in months.

A table including clinical and molecular information would be useful, along the lines of Table 1 in this article. https://www.sciencedirect.com/science/article/pii/S000292971830140X?via%3Dihub

response: we have described phenotype/ clinical features in tables 1-2-3 and the genomic findings in table 4. we are somewhat limited with the formatting with the size of tables so we divided this way.

Line 336: The acronym “FADS” should be explained.

response: the acronym was spelled out (fetal akinesia deformation sequence)